# Jane Austen: The Musician as Author

Gillian Dooley

Department of English, College of Humanities, Arts and Social Sciences, Flinders University, Adelaide 5000, Australia; gillian.dooley@flinders.edu.au

**Abstract:** Jane Austen was a practising musician, and my intention in this paper is to investigate the significance of that fact for her writing practice. Beginning with the comparison between Elizabeth and Mary Bennet in *Pride and Prejudice*, I will consider contemporary attitudes to virtuosity and aesthetics in an attempt to understand the implications in Austen's fiction of the distinction between 'playing well' and 'being listened to with pleasure'. My recently completed project of cataloguing in detail each piece of playable music in the Austen Family Music Books facilitates the study of Austen's personal musical taste in the context of her extended family and, more broadly, of English musical culture in the late Georgian era. I attempt to bring together Austen the musician with Austen the writer, both in her knowledge of the musical repertoire of the time and the language of music more generally.

**Keywords:** Jane Austen; music; musician; Austen Family; *Pride and Prejudice*

## Introduction

Early in *Pride and Prejudice*, two of the Bennet sisters play the piano at an evening party. Elizabeth plays first, and then Mary. Elizabeth 'had been listened to with much more pleasure' than Mary, 'though not playing half so well' (J. Austen 1932, p. 25). This is clearly an indication of the difference in the characters, Mary's vanity and affectation are explicitly contrasted with Elizabeth's relaxed and sociable nature. However, considering that Jane Austen was herself a musician and a discerning listener to the performance of others, I am interested in understanding the broader aesthetic implications of statements like these.

Austen's music collection, including nearly 200 manuscript items copied in her own hand as well as printed music, provides a framework in which to study the music she was drawn to and that she is likely to have played and sung. Moreover, although none of Austen's characters are writers like her, several share her amateur musical activities, and it seems possible that her observations on the various capacities and performances of her musical characters might give some insight into Austen's own attitudes and practices as a writer. Music, especially singing, was seen at the time (and indeed still is seen) as a rhetorical art. Rhythm, cadence, voice, and tonality are all important both in writing and in music, and Austen's prose is well known for its mastery of all these elements. How might these factors be brought together to illuminate Austen the writer alongside Austen the musician?

## Literature Review

This article is part of a larger project on music and Jane Austen, which has already produced several articles in recent years. A book is in preparation. In a sense, this essay is a companion piece to an article published in *Persuasions*, titled '"There Is No Understanding a Word of It": Musical Taste and Italian Vocal Music in Austen's Musical and Literary World' (Dooley 2022). In that article, I look in more detail at the question of taste, with particular focus on *Emma*. For Austen's contemporaries, ideas about virtuosity were very much linked to discussions of aesthetic taste, so these two articles share a broad frame of

reference. However, this article concentrates more on what Austen would call 'execution', rather than musical taste.

My project builds on the work of several scholars. The first book to consider Austen's rhetoric alongside musical forms was Robert Wallace's (1983) *Jane Austen and Mozart*, which compares Austen's fiction with Mozart's piano concertos, and includes two informative appendixes about the music in Austen's surviving collection of songs and piano works. However, Wallace only had access to one half of the collection when he wrote his book.

More recently, G. Wood's (2010) book *Romanticism and Music Culture in Britain, 1770–1840* includes a chapter on 'Austen's Accomplishment', discussing ideas of virtue versus virtuosity and how Austen manages to bring these two apparently opposing notions into harmony. In the book's introduction, he discusses the

> discursive *cordon sanitaire* designed to separate the luxury, effeminacy, empty sociability, and mechanical display associated with music from the nascent values of inferiority, sincerity, and sublimity that would define Romantic literary culture . . . [V]irtuosity served for Romantic writers as a composite bogey embodying both the aristocratic tradition of the amateur against which their own professionalised practices would be drawn. (G. Wood 2010, p. 6)

This tension between empty virtuosity and Romantic sincerity is implicit in many of the musical references in Austen's novels, especially *Pride and Prejudice*, where musical performance is seen, by some of the characters at least, as a field for competition. Jeffrey Nigro and Andrea Cawelti discuss 'Divas in the Drawing Room', the historical background to Austen's competitive performers in *Emma*, who embodied the different sides of the debate: 'The music of the Italian opera made its way to English drawing rooms in Austen's day as did stories of the clashing temperaments of singers'(Nigro and Cawelti 2016). They concede, though, that this rivalry, in Austen's novels, becomes transmuted into 'an encouragement to wholeness and harmony' and that Emma's 'diva' status is tempered in the course of the novel into someone who is 'part of an ensemble, that is, a community' and that her friends are 'her colleagues and not merely her audience.' (ibid.)

I too am interested to see how her own attitudes to musical and artistic virtuosity, as far as they can be inferred from her letters, her music collection, and various memoirs written by her relatives, are manifested in her writing. I believe this is evident not only at the level of theme and character but in rhetorical terms: Musical influences can be traced in her prose and the structure of her novels. Could she have learned her craft not only from Samuel Johnson, Frances Burney, and Samuel Richardson but also from Charles Dibdin, Stephen Storace, and Giovanni Paisiello?

### *Pride and Prejudice*—Display versus Pleasure

Ambivalence towards virtuosic music, or music used merely for display, is dramatised in several ways in Austen's novels, nowhere as clearly as in the contrast between the two musical Bennet sisters in *Pride and Prejudice*. Early in the novel, at an evening party at the home of the Lucas family, Elizabeth's friend Charlotte, the eldest daughter, asks her to perform at the piano. Elizabeth responds,

> 'You are a very strange creature by way of a friend!—always wanting me to play and sing before anybody and everybody!—If my vanity had taken a musical turn, you would have been invaluable, but as it is, I would really rather not sit down before those who must be in the habit of hearing the very best performers.' (J. Austen 1932, p. 24)

The immediate context of this exchange is that Elizabeth is becoming aware of Darcy's interest in her: 'He began to wish to know more of her, and as a step towards conversing with her himself, attended to her conversation with others. His doing so drew her notice' (ibid.). She does not let his silent observation pass unremarked: 'If I do not begin by being impertinent myself, I shall soon grow afraid of him' (ibid.). Therefore, when Charlotte asks her to play, she is already combatting self-consciousness. Her protest about her vanity

not 'taking a musical turn' is no doubt intended to be heard not only by Darcy but also by the Bingley sisters, who she suspects must be used to hearing 'the very best performers'. Having excused herself in advance for her lack of virtuosity, she performs:

> Her performance was pleasing, though by no means capital. After a song or two, and before she could reply to the entreaties of several that she would sing again, she was eagerly succeeded at the instrument by her sister Mary, who having, in consequence of being the only plain one in the family, worked hard for knowledge and accomplishments, was always impatient for display. (J. Austen 1932, p. 25)

The music Austen would have imagined Elizabeth playing and singing that evening is probably the kind of music she played and sang herself. It would be perhaps a sentimental song by William Shield or James Hook, or the Duchess of Devonshire's 'Silent Sorrow', or even a French song, like Austen's favourite, 'Que j'aime à voir les hirondelles', by François Devienne, followed by a more cheerful Scottish ballad. There would be nothing too technically challenging, a pleasant melody with a simple (possibly improvised) accompaniment, and played, probably, from memory. Mary, on the other hand, chooses to play 'a long concerto', a work for solo piano of some complexity which, even leaving aside the standard of her performance, is out of place for an evening party rather than a concert[1]. Mary would have to have prepared for a performance like this either by memorising a long and complex work or by bringing the sheet music with her in anticipation of having the opportunity to perform.

Lady Catherine de Bourgh, during Elizabeth's visit to Rosings, remarks that 'Miss Bennet would not play at all amiss, if she practised more, and could have the advantage of a London master. She has a very good notion of fingering, though her taste is not equal to Anne's' (J. Austen 1932, p. 176). She continues her comments 'on Elizabeth's performance, mixing with them many instructions on execution and taste' (ibid.). Lady Catherine has already declared herself an expert:

> There are few people in England, I suppose, who have more true enjoyment of music than myself, or a better natural taste. If I had ever learnt, I should have been a great proficient. (ibid., p. 173)

Austen's heavy irony at the expense of the great lady is no doubt the main target in much of this chapter, although there are several other important threads, including Darcy's silent observation of Elizabeth's animated conversation with Colonel Fitzwilliam. However, Lady Catherine's grudging praise of Elizabeth's musicianship provides a little more insight into her musical skills and, perhaps, her choice of music. The concept of taste recurs in all these situations. Did Elizabeth perhaps sing something a little more vulgar than would be normal in the aristocratic drawing room: A comic song by Charles Dibdin, for example? Or possibly she played only English music and avoided the Italian songs which were more in vogue.

The earlier scene at the Lucas's party provides further expansion on the contrast between Mary and Elizabeth, and perhaps clues as to what some of the key terms might have meant to Austen:

> Mary had neither genius nor taste; and though vanity had given her application, it had given her likewise a pedantic air and conceited manner, which would have injured a higher degree of excellence than she had reached. Elizabeth, easy and unaffected, had been listened to with much more pleasure, though not playing half so well. (J. Austen 1932, p. 25)

By 'genius', Austen presumably means natural talent or aptitude which translates to apparent ease in performance—something which cannot be learned. Elizabeth has this trick of being 'easy and unaffected'—an important attribute in a performer. One often hears a musician praised for 'making it sound easy' despite the technical difficulties of the music they are playing. Without this capacity, the audience is unable to relax and enjoy the music without anxiety. Although performance nerves are common even among the greatest

musicians, a performer who is tense and anxious to impress is not able to communicate their joy and feeling for the music, whatever its level of difficulty. A performer needs to be self-confident without being conceited.

It is interesting to consider exactly what 'not playing well' might mean in this context. Was Elizabeth playing wrong notes? Or was she playing simpler music that did not require the level of virtuosity or accomplishment that Mary aspired to? The implication is that her technical skills are inferior but even if she did miss some notes and played easier music than Mary, she was engaging in an act of communication with her audience, who asked her for more, while they were unimpressed by the 'impatience for display' with which Mary approached her performance.

Italian music, which in those times practically epitomised virtuosic music, plays only a very small explicit role in *Pride and Prejudice*: 'After playing some Italian songs, Miss Bingley varied the charm by a lively Scotch air' (J. Austen 1932, p. 51). This is one of several references in the novels to Scottish music being offered to an audience that is possibly bored by the more serious, or showy, music that they have been hearing: 'Mary, at the end of a long concerto, was glad to purchase praise and gratitude by Scotch and Irish airs, at the request of her younger sisters' (ibid., p. 25). Unlike Elizabeth, who had been asked for more, poor Mary has to appease her audience by playing what, perhaps, she regarded as less worthwhile music.

Aside from *Northanger Abbey*, each of the other completed novels has one or more musical characters. For each of these characters, music has a different meaning. The debate between 'virtue and virtuosity' is not completely absent elsewhere, but in the other novels, it is tempered by other factors. The stark contrast between giving pleasure and 'playing well' is illustrated most clearly in the Bennet sisters.

**The Meaning of Music**

In mid-2020, Australian concert pianist and writer Anna Goldsworthy wrote from the depths of pandemic-struck Adelaide about the meaning of music. For musicians, the inability to do what they had spent their lives training to do, to play well *and* give pleasure to others, had struck at the heart of their being:

> The music is still there . . . But somehow, in the absence of an audience—or the expectation of an audience, tomorrow or next week—it becomes harder to pick that instrument up. I wondered if this revealed an unfortunate character flaw: That showing off had, after all, been the entire point. But . . . [p]erforming is an act of communion: With the composer, with your colleagues, but also—critically— with your audience, which almost wills the experience into being. It offers a mode of connection that can feel telepathic. (Goldsworthy 2020)

I do not believe that the meaning of music was fundamentally different in Austen's time from ours. Playing well, to her, would have meant what it means to modern musicians, whether amateur or professional. One crucial difference is that in Austen's time, music could only be heard at all if there was a musician physically present. Those who enjoyed music were, therefore, perhaps, somewhat more likely to be active musicians as otherwise, they had to depend on circumstances to hear music at all. In any case, genuine musicians are communicators. They play as well as they can, according to their level of skill, having chosen music suitable for the audience and the occasion. However, the soulless virtuoso, the 'bogey' of the Romantic movement (G. Wood 2010, p. 6), seeks only to impress.

One evening as I was struggling with ideas for this article, I heard the great Italian virtuoso Salvatore Accardo on the radio playing a violin concerto by Pietro Locatelli (1695–1764). The work was almost comically vacuous while being virtually impossible to play well. It had no intrinsic beauty or even harmonic variation, tonic and dominant chords alternated underneath rapid-fire passagework at the highest extent of the violin's range which even Accardo could not play in tune. The only point of the piece seemed to be to show off how high and fast the violin could be played. Accardo was featured on this program, and his playing elsewhere was beautiful and often moving.

This experience reminded me of a concert I attended many years ago by a well-known American singer. She sang faultless coloratura, works by Handel and other Baroque composers whose work I love. However, I was unmoved and even insulted by her performance. I felt that she had no wish to communicate with us but only to dazzle us, whom (she somehow conveyed) she regarded as a provincial and unimportant audience.

The former, it seems to me, is an example of *music* that is emptily virtuosic, and the latter an example of a virtuosic *performer*, in the worst sense of the word. However, as Kathryn Libin points out, summarising the opinion of Charles Burney, 'taste and execution are seldom fully united in one player. When they are, that player is a virtuoso' (Libin 2000). There are few of these genuine virtuosi in Austen's novels: Jane Fairfax is one, Marianne Dashwood perhaps another. Neither Mary nor Elizabeth Bennet fit the definition.

Immediately before Lady Catherine's overbearing advice to Elizabeth about not practising enough, Darcy and Elizabeth have been discussing piano practice as an analogy of social skills. Darcy says he does not have 'the talent which some people possess . . . of conversing easily with those I have never seen before' (J. Austen 1932, p. 175). He implies that his reserve, or shyness, is something inherent within him that he cannot change. Elizabeth responds pointedly:

> 'My fingers . . . do not move over this instrument in the masterly manner which I see so many women do. They have not the same force or rapidity and do not produce the same expression. But then I have always supposed it to be my own fault—because I would not take the trouble of practising. It is not that I do not believe *my* fingers as capable as any other woman's of superior execution.' (ibid.)

Her point is obvious: If he 'practised' he would be able to learn the 'talent' of talking to strangers if he wanted to, just as she could become a virtuoso pianist if she applied herself. However, in saying that she had chosen not to take the trouble to practise, she is surely allowing that he is free to make the equivalent choice for himself. Whether or not this is what she intends to say, it seems to be how he chooses to interpret her remark. He smiles and says,

> 'You are perfectly right. You have employed your time much better. No one admitted to the privilege of hearing you can think anything wanting. We neither of us perform to strangers.' (J. Austen 1932, pp. 175–76)

In a 2017 article, I discussed my puzzlement at this remark of Darcy's, and the conversation with Elizabeth that precedes it (Dooley 2017a). However, looked at from the point of view of amateur versus professional musicianship, it makes sense that he would approve of what he seems to regard as Elizabeth's anti-virtuosic stance. To hear Elizabeth play, he points out, one must be 'admitted to the privilege of hearing' her. She is not a public performer: A professional available for hire to strangers, a mere virtuoso with all the shallow attractions of technical skill. Her musicianship, like his conversation, is for communication among friends and family, not for 'performing to strangers'. Austen herself, according to her niece Caroline, 'was never induced (as I have heard) to play in company' (C. Austen 2002a, p. 170).

### Austen's Own Attitudes to Musical and Artistic Virtuosity

Much of the information that we have about Austen's musical activities comes from her niece Caroline, the younger daughter of her brother James, who was only twelve when Austen died. Caroline wrote in 1867 that the music her Aunt Jane played and sang for her in her childhood included 'very pretty tunes, *I* thought . . . but the music (for I knew the books well in after years) would now be thought disgracefully easy' (C. Austen 2002a, p. 171). Robert Wallace quotes this statement and goes on,

> Caroline's evaluation speaks for her era as well as herself. By 1867 the powerful and vehement music not only of Beethoven but of such Romantic successors as Berlioz, Schumann and Liszt had, in most people's minds, rendered 'disgracefully

easy' the music not only of the 'classical' composers Austen had played but even that of Haydn and Mozart themselves. (Wallace 1983, pp. 262–63)

Austen herself expressed a view that reflected the attitudes of her own time when in 1811, she reported on a musical evening at the home of her brother Henry and his wife (their cousin) Eliza:

The Music was extremely good. . . . There was one female singer, a short Miss Davis all in blue, bringing up for the Public Line, whose voice was said to be very fine indeed and all the Performers gave great satisfaction by doing what they were paid for and giving themselves no airs. (J. Austen 1995b, p. 183)

Professional musicians at that time 'were regarded as artisans or tradespeople and not accepted as the equals of those for whom they performed' (Dooley et al. 2018). The Romantic movement eventually improved the status of artists, including professional musicians, as the nineteenth century proceeded. Amateur musicians were quite different: Like Mr Darcy, they were more likely to be proud of not 'performing to strangers'.

### Music and Rhetoric

Austen habitually read her own works aloud to her family and friends. Her niece, Anna Lefroy (Caroline's older sister), recounts being told in later years

that one of her earliest novels (Pride and Prejudice) was read aloud (in M.S., of course) in the Parsonage at Dean, whilst I was in the room, and not expected to listen.—Listen however I did, with so much interest, and with so much talk afterwards about 'Jane and Elizabeth' that it was resolved, for prudence sake, to read no more of the story aloud in my hearing. (Lefroy 2002, p. 158)

Anna was four years old in 1797 when *First Impressions* was finished. Austen must have been an engaging reader to catch the attention of so young a child, who was expected not to be interested in listening.

Caroline also recalled that Austen was 'considered to read aloud remarkably well. I did not often hear her, but *once* I knew her to take up a volume of Evelina and read a few pages . . . I thought it was like a play. She had a very good *speaking* voice.' (C. Austen 2002a, p. 174). Good writing is easy and natural to read aloud, I find that reading Austen's prose aloud is always a great pleasure. It is always a good test of one's own writing to read it aloud, even if it is intended only for print publication. It is not surprising to hear that she was a good speaker and read well.

She herself noted the importance of reading aloud in her letters to Cassandra of 29 January and 4 February 1813, when her copy of *Pride and Prejudice* had just arrived. A visitor, Miss Benn, 'dined with us on the very day of the Books coming, and in the even^g we set fairly at it and read half the 1st vol. to her', without letting on who the author was. Miss Benn 'was amused, poor soul!' (J. Austen 1995c, p. 201). On the second occasion, Austen was not so pleased, which she thought 'must be attributed to my Mother's too rapid way of getting on—and tho' she perfectly understands the Characters herself, she cannot speak as they ought' (J. Austen 1995d, p. 203).

In 1814, she attended a performance of Thomas Arne's *Artaxerxes*, with the virtuosic soprano Catherine Stephens in the role of Mandane. She wrote to Cassandra deprecating Stephens' acting skills and expressing indifference to her singing, which seems to confirm her awareness of the importance of communication in performing (J. Austen 1995a, p. 261). According to Robert Toft, singing at this period was regarded as a kind of acting, where 'singers were taught to imitate real life, and by imagining that they actually were the very person they were representing . . . they became animated with the passion to be expressed' (Toft 2000, p. 15). This is the antithesis of empty virtuosity, and although Stephens was much admired by many, Austen was not moved by her performance.

Being a good reader requires similar skills to being a good musician, perhaps especially a singer. As Toft writes, 'Several writers from the period declare that singing should be based directly on speaking and that singers should use the orator as a model.' (ibid.)

However, there are similarities between music and prose, even for an instrumentalist. In an interview with pianist and writer, Anna Goldsworthy, I asked her about the resonances she sees between writing and playing the piano:

> [T]here is that larger structural formal thing, and I think on the more sort of microscopic level there are . . . musical concerns, and they are rhythm, they're cadence, they are modulation, modulating from one paragraph to the next, coming in with a new tone of voice. . . . And then, by extension, if I'm working with a piece of Beethoven or Schubert or whatever, I think a lot of the laws of grammar and punctuation come into it. . . . And I'll often say to [my piano students], well is that [rest] a comma, is that a full stop, is that a semi-colon, is that a colon? Because it could be all of those things. (Goldsworthy and Dooley 2017)

> In John Wiltshire's innovative book, *The Hidden Jane Austen*, he points out that she

> gets her results and controls her meanings, largely through the precise exercise of syntax: Grammatical construction, punctuation, emphasis, and rhythm. . . . [w]hen her characters speak, she frequently employs a distinct register of syntactic markers as signals for the informality of conversational utterance. . . . Among such mimetic markers were dashes of varying lengths, exclamation marks, incomplete sentences, italics and repetition. (Wiltshire 2014, p. 6)

These devices are used not only in spoken conversation but in inner monologues, like Fanny Price's agonised reaction in *Mansfield Park* to Edmund's letter confiding in her his wishes and fears regarding Mary Crawford: 'Oh! write, write. Finish it at once. Let there be an end of this suspense. Fix, commit, condemn yourself' (J. Austen 1934, p. 424). As Wiltshire writes, in her later novels, 'the dash is put to even more daring uses' (Wiltshire 2014, p. 8). The dash allows for incompleteness, leaving room for implications that the character might wish to evade or at least leave unsaid, in the same way that pauses in music allow the listener to hear and digest what has been heard and anticipate what is to come:

> Largely dispensing with the poetic language of feeling, Austen can nevertheless lodge emotion in other discursive gaps besides dashes—in the silences and pauses that the prose dramatises in what the narrator understates, and in what she simply elides by shifting the reader's focus. (ibid., p. 9)

Setting these insights into Austen's rhetoric beside Goldsworthy's comments about the grammatical status of rests and pauses in music allows us to see what the two arts share, succinctly summarised in novelist and essayist Charlotte Wood's book *The Luminous Solution*, where she formulated, with writing colleagues, the essential elements of a compelling sentence: 'Clarity, authority, energy, musicality and flair' (C. Wood 2021, p. 126). Wood tells of her growing understanding, in parallel to her grasp of what made prose compelling, of what she was actually responding to when listening to music.

> Over time, I grew to understand how little I'd noticed before and that what was really affecting me so bodily in a favourite song had surprisingly little to do with the surface melody of a piece. Much more, I was pulled into a song by the underlying attraction of its harmonic and rhythmic structures and counter melodies. (ibid., p. 129.)

Wood learned to appreciate this depth in music from her husband, who was a musician. Austen's understanding of music from the inside cannot have but affected the rhythms, harmonies and cadences of her prose.

Having spent several years recently immersed in the Austen Family Music Books (2015) in the process of researching and cataloguing each individual item, I have now ventured beyond the purely literary critique in my earlier work and sought links between the music Austen played and sang and her writing[2]. I have written on what I saw as possibly suggestive links between a ballad setting by Tommaso Giordani and *Sense and Sensibility*, including the shared rhetoric of the two works: Repetition of brief exclamations for dramatic effect, for example (Dooley 2018). For the JASNA AGM in 2020, I explored the

music she is likely to have been playing and singing during her teenage years when she was writing her Volumes the First, Second, and Third (Dooley 2020b). In particular, I noted the number of songs by Charles Dibdin that she copied into her manuscript books. Many of these songs are comic and contain the kind of hyperbole, surprises, and comic reversals to be found in her early work. I believe that the reputation Austen has among critics as a virtuosic writer can be traced to her early experiments and mature mastery of rhetorical techniques like these. Perhaps it might be said that she grew out of the showy virtuosity of the early writings into a more mature style where writing expertise becomes employed in the cause of a more genuine communication with her readers.

**European Music in Austen's Collection**

The debate about virtuosity was often framed as one between European (especially Italian) virtuosity and English virtue. Austen had both music from the British Isles, in English, and European music, usually in French or Italian, in her collection.

The clearest and earliest 'foreign' influence probably comes from her cousin Eliza Hancock. Eliza moved to Paris in 1779 with her mother and married Jean-François Capot de Feuillide in 1781. She often visited the Austen family in Steventon during Jane's childhood and teenage years. Although she was fourteen years older than Jane, they were close. Eliza played both the piano and the harp. Following the execution of her first husband in Paris in 1794, Eliza married Jane's favourite brother, Henry, in 1797.

There is evidence of Eliza sharing music with her young cousin. Towards the front of one of her own manuscript books (titled 'Juvenile songs and lessons for young beginners who don't know enough to practise'), Austen carefully copied two overtures to French operas, arranged for harp or keyboard, from Eliza's printed music. Most of the other items in that book are English or Scottish keyboard pieces.

The other early manuscript book is titled 'Songs and Duetts' and contains 37 songs, most of which are in English. The seven French songs in the album are not especially virtuosic and not more musically complex than the English-language ones, but there is often a certain sophistication about their lyrics and their musical settings, which seems to set them apart from many of the English songs.

Perhaps the most trenchant is an unattributed song titled 'Plus ne veux jamais m'engager'. The words are printed in a later volume of French poems and attributed to 'Heurtier', but I have not been able to find this song anywhere else, and neither can I discover anything about the poet Heurtier. The singer here is thoroughly disillusioned and wishes to have nothing further to do with love. The first verse translates:

> I will never involve myself again—
> Such is my fancy.
> Because every lover is flighty,
> To really love is madness.
> Faith is no more, nor integrity;
> It's a strange thing—
> People love only in vanity,
> That people change is vanity. (*Air ['Plus ne veux jamais m'engager']* n.d.)

The melody is disarmingly simple, but as a setting of these bitter words, it creates a bitingly cynical statement.

Another piece that is startling in its emotional intensity is rather deceptively titled 'Air du Marquis de Tulipano'. The Marquis of Tulipano was a 1789 'Opera Bouffon', a comic 'parody opera' put together by J.A. Gourbillon based on an Italian opera of Giovanni Paisiello, one of the most renowned composers of the time, who worked in Paris for a short time in the 1780s. However, there is nothing remotely amusing about this song, 'Je croyais ma belle':

> I thought, my love,
> I could taste forever such perfect pleasures,
> (Alas, my love)
> In being faithful to those perfect pleasures.
> To cap my troubles,
> Sleep escapes me;
> I spend the night
> Cursing my fate;
> In vain I pretend
> To flee your attractions;
> Ever more lamenting
> I find myself, alas. (*Air du Marquis de Tulipano* n.d.)

To add to the intensity of this beautiful aria, it is in a minor key, unusual in this period: The vast majority of the items in the Austen music collection are in major keys. The words 'Hélas, ma belle!' ('Alas, my love!') are repeated several times in a sequence of short melodic phrases, forming on just those three words a world of emotional depth. There is nothing quite so boldly passionate as these two songs among the English-language pieces, although there is considerable variety in the musical and lyrical qualities of the English songs. Some rely heavily on poetic clichés of the time, while others are more effective in their directness and humour.

Even the most charming and humorous of the French songs is knowing in its own way. 'La danse n'est pas ce que j'aime' comes from the 1784 opera *Richard the Lion-Heart* by the eminent Belgian-born composer André Grétry. The minor character who sings it is a 16-year-old boy (sung by a soprano in the opera), and he is describing his girlfriend to another character:

> It isn't the dance that I love,
> But the daughter of Nicolas;
> When I hold her in my arms,
> Then my pleasure is extreme.
> I press her against me. (*De Richard Coeur de Lion* n.d.)

The young man goes on to describe how he and his lover (who is fifteen) try to evade the notice of her mother and talk to each other 'tout bas', very softly. The phrase 'tout bas' is repeated softly four times, imitating the secretive conversations between the lovers, and then the young man breaks out, at full voice with 'Ah, que je vous plains!' (Oh, how I pity you!), which is then repeated at a higher pitch—'Vous ne la verrez pas!' (You will not see her).

Another song, which begins 'Laisse là sur l'herbette ton chien et ta houlette', is a dialogue between an over-confident lothario named Lucas and a feisty shepherdess who roundly rejects his advances. A print version of this song is in the Bodleian Library, and the published title is 'Le Refus'—the Refusal. The young woman does not merely say 'no' once but in the course of the three verses, repeats the word 'no' sixty-six times, on emphatic descending scales, repeated with embellishments, showing increasing impatience with the insensitive assumptions of this uncomprehending, entitled lout. It is possible to hear an echo of these songs in such scenes as Mr Collins' proposal to Elizabeth in *Pride and Prejudice* or Mr Elton's advances to Emma.

In these two songs, humour and sexual innuendo are to the fore, while in the first two French songs I discussed, there is more intensity of emotion conveyed both by lyrics and music, perhaps echoed in the suppressed suffering of Colonel Brandon in *Sense and Sensibility*, or indeed of Edmund rejected by Mary Crawford in *Mansfield Park*.

There is sufficient French music in Austen's music books to make up the core of a concert program for voice and harp. There is less Italian music that we can definitely connect with Austen herself, although her sister-in-law Elizabeth née Bridges, married to Edward Austen Knight, both owned and copied into her own manuscript books a greater

proportion of Italian arias, many of which Austen would have been familiar with. I have discussed the three Italian songs in Austen's own hand in a 2022 article in *Persuasions* (Dooley 2022). One is a well-known Venetian folksong, 'La Biondina in Gondoletta'. Another is a humorous song about a chimney sweep that she might have entertained her young relatives with. The third is a charming, but not difficult, aria titled 'Oh Giovinetti', which I have not been able to identify.

In addition to these French and Italian songs, among Austen's own music, there is evidence of an enduring interest in cultural influences beyond the borders of southern England, including three German songs, a 'Polonese Russe', an 'Egyptian Love Song', and a 'Hindoo Girl's Song', as well as songs in other languages and instrumental music from abroad. However, the foreign music is, on the whole, not the empty virtuosic music of the stereotypes at the time. On the contrary, the songs are well within the abilities of a good amateur singer, requiring not vocal dexterity but directness, sincerity, and observation of natural speech rhythms.

**Political Dimensions**

Caroline Austen observed that

my Aunt must have been a young woman, able to *think*, at the time of the French Revolution and the long disastrous chapter then begun, was closed by the battle of Waterloo, two years before her death—anyone *might* naturally desire to know what part such a mind as her's had taken in the great strifes of war and policy which so disquieted Europe for more than 20 years—and yet, it was a question that had never before presented itself to me—and tho' I have now retraced my steps on *this* track, I have found absolutely nothing! (C. Austen 2002a, p. 173)

Caroline knew the music collection but did not make the connection between any thoughts Austen might have had about 'public events' (ibid.) and some of the French songs that she collected.

'Pauvre Jacques' may have been written by Queen Marie Antoinette or one of her companions. It dates from very shortly before the Revolution and is said to be based on the lament of a young Swiss woman employed in the Queen's dairy, who is sorely missing her lover and her home. 'Chanson Béarnoise', on the other hand, comes from after the French Revolution and is a Royalist ballad attacking the treachery of those who have imprisoned Louis XVI and his queen. This does not appear to have been available in England but was freely available on the streets of Paris[3].

Also included in the same book is 'Captivity', by the English-Italian composer Stephen Storace, one of several English songs composed about the plight of Queen Marie Antoinette, imprisoned and awaiting her fate at the hands of the revolutionary government, as well as 'Queen Mary's Lamentation', a corresponding piece which dramatises the imprisonment of Mary Stuart, which confirms Caroline's memory of Austen's loyalty to the Stuart cause (C. Austen 2002a, p. 173).

On the other hand, there is 'The Marseilles March', an early version of the Revolutionary hymn 'The Marseillaise', now, of course, the French national anthem. Although when the song was composed by Rouget de Lisle in 1792 France and England were not at war, evidence suggests that Austen copied the melody and six of the original bloodthirsty verses in French no earlier than 1794, after her cousin's husband had been executed by the revolutionary government and after the outbreak of war between Britain and France in February 1793.

These songs, all in their way, reflect the ferment of the dramatic events in France during the last years of the eighteenth century and show that the young Jane Austen was certainly aware of what was happening there. It is probable that most of these songs were shared with her by her cousin Eliza. In all of these songs, there is drama and a narrative, and a point of view being presented, whether it is French revolutionaries baying for blood or Marie Antoinette quaking in horror at her husband's death and dreading her own fate. There is little evidence of a definite bias, at least as far as the French situation is concerned.

There seems to be more interest in the performance of a dramatic point of view. None of these songs is in the least virtuosic. They are all set to dignified, simple melodies which do not get in the way of the words being heard and understood.

Decades after Austen's death, Caroline remembers her often singing and playing another song in French, a romance beginning 'Que j'aime à voir les hirondelles' (C. Austen 2002b, p. 193). This song does not appear in any of Austen's surviving manuscript books but is to be found in a book which had belonged to Eliza. Composed in 1788 by François Devienne, it also deals with the trope of captivity and death: A swallow captured by a cruel child and kept from its faithful lover will die 'd'ennui, de douleur et d'amour', of languor, sorrow, and love. The lyrics come from the pastoral novel *Estelle et Nemorin* by Jean-Pierre Claris de Florian (1755–1794). The melody is well-crafted and suits the pathos and drama of the words.

### Other Music in the Collections

This overview of the vocal music in the Austen family's music collection leaves out a large amount of other music set for piano or harp, much of which originated from Europe or was written by some of the many European composers working in England during the period. Many of these works are discussed in Appendix One of Robert Wallace's *Jane Austen and Mozart* (Wallace 1983). Austen's own music books, especially the later ones, show an eclectic mixture of music of the time, folk songs, theatre songs, opera, and keyboard music of various levels of complexity. Little of it is still in the classical music repertoire, apart from a couple of small unattributed pieces by Mozart, and several pieces by Handel and Haydn. Little of it requires an extended vocal technique or elaborate accompaniment. What the songs have in common is that the lyrics tell a story, embody a character, sometimes in a comic vein, or dramatise a situation in relatively simple, uncluttered musical settings.

### Conclusions: The Writer as Virtuoso

In 1924, Virginia Woolf wrote of Austen that 'of all great writers, she is the most difficult to catch in the act of greatness (Woolf 1924). As John Wiltshire writes, Austen does not often use 'the poetic language of feeling', and 'when she uses metaphorical language, it is usually to register its banality' (Wiltshire 2014, pp. 6, 8). Perhaps poetic language and metaphor are the 'acts of greatness' or virtuosity that many readers expect to be able to see on the page, and so they tend to look through the prose as if it were a transparent frame for a picture of women in empire line frocks and handsome men in cutaway coats and white breeches making eyes at each other. However, something has to account for her continuing power, something that her contemporary novelists lack.

It was perhaps not until I had been reading and studying her for several decades that I realised just how important the rhythm and structure of her sentences were, how beautifully balanced they are, how often they contain a surprise. How virtuosic and hard to pin down is her dazzling deployment of free indirect style. Although increasing attention has been paid to music as a theme in Austen's novels and also to the contents of her music collection, little has been written about the musical nature of Austen's prose since Robert Wallace's book 1983 comparing the 'classical equilibrium' of Austen with that of Mozart.

A few years ago, I gave a talk at my local Jane Austen Club about how to read for point of view, rather than arguing about which characters they like and dislike. I am proud to say they have come to this idea with great enthusiasm and are now listening for the interplay of voices in the narratives (Dooley 2017b). Like Charlotte Wood, a virtuoso novelist of the present day, discovering that her enjoyment of music had little to do 'with the surface melody of a piece' and much more to do with the 'underlying attraction of its harmonic and rhythmic structures and counter melodies,' (C. Wood 2021, p. 129) these intelligent but non-expert readers were learning to look at, and listen to, the words on the page to appreciate Austen's artistry and skill. As Wood writes,

> Attention to its musicality shows a sentence as capable of much more than melodiousness—rhythm alone can argue, preach, threaten, jolt or suffocate as easily as it can soothe or invigorate. The energy in a sentence might come from its voice, from mystery, from compression, reversal or surprise. (ibid., pp. 127–28)

Austen, sitting at the piano for an hour each morning, or even playing simple country dances for her nieces and nephews, was regularly embodying this rhythm and energy, and then when she sat at her writing desk, it is only natural that the narrative energy of the music she played should enter and inflect her prose.

**Funding:** This research received no external funding.

**Institutional Review Board Statement:** Not applicable.

**Informed Consent Statement:** Not applicable.

**Data Availability Statement:** Not applicable.

**Conflicts of Interest:** The author declares no conflict of interest.

## Notes

[1] An example of a concerto for keyboard in Austen's music collection is by William Evance. It could be performed by a piano soloist but also has string parts that double the keyboard part. It is not like the concertos by Mozart, which have a piano part completely separate from, and complementary to, the orchestra—see https://www-lib.soton.ac.uk/uhtbin/cgisirsi/x/0/0/57/5/3?searchdata1=1786645%7BCKEY%7D&searchfield1=GENERAL%5ESUBJECT%5EGENERAL%5E%5E&user_id=WEBSERVER (accessed on 10 June 2022).

[2] My first publication on music in Austen's fiction was 'Musicianship and Morality in the Novels of Jane Austen' (Dooley 2010).

[3] A detailed discussion of these French songs in the context of historical events and Austen's life can be found in Gillian Dooley (2020a).

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
