# Peer review of "Jane Austen: The Musician as Author"

_humanities, doi:10.3390/h11030073_

Round 1

Reviewer 1 Report

This article addresses the impact of Jane Austen’s experience as a musician on her writing. The article touches on music in the life and in the novels exploring performance, structure and rhetoric. It reads the concept of virtuosity and communication in performance and then analyses some of the music personally copied by Austen both as a feature of her skill and as a rebuttal to her ignorance of historical events. The analysis of the Bennet sisters is clear and well-targeted and the discussion of the music collection adds further value. This latter is of particular interest in the life and legacy aims of this special issue of the journal.

The article is well-written and the discussion of Pride and Prejudice and the music collection is largely compelling; the inclusion of modern musical commentators less so. The personal intrusion of the author of the article distracts from the line of argument; personal reference is clearly appropriate to the research into Austen’s music but elsewhere this becomes more like a conference paper than a published article. This can easily be corrected to create a more authoritative tone.

The opening of the article needs to be strengthened and the planned structure made more open with signals of the movement of the argument. The quotation from Wood is particularly undigested and could be better integrated into the discussion. The penultimate sentence on page 1 is very strong but the opening (lines 1–37) rather tentative; there is no need to apologise for building on the work of others even though they must be referenced. Lines 38–40 can make a strong statement of intent. The abstract could be made more purposeful and descriptive of the actual process followed.

Reading further, it is almost as if there are two articles here: one about performance and the other about the music copied in Austen’s lifetime. The discussion of the Bennets whets the appetite for an exploration of Marianne Dashwood, Jane Fairfax, Emma and Anne Elliott that never comes. Reference to the article about ‘class’ (Dooley, Moffat and Wiltshire) suggests that the topic within the novels has been covered elsewhere. Perhaps, therefore, the author of this article should make special use of their unique access to the music. There is enough here for this reader to want to know more. The article would be strengthened itself by more concise Bennet material as an introduction to the detail of Austen’s own music (eg her piano-playing at Steventon and Chawton and the impact of living in Bath/Southampton with rented piano). This would supplement the original contribution being made by the article author’s work on the collection. Is it possible to speculate on the life experience of music influencing the later novels or the rewriting of the earlier ones? This would reinforce the suggestion of a rhetorical impact from the music circulating in Austen’s life. The sections on reading out loud as performance can be supplemented from Austen’s letters; she read P&P out loud and was unhappy about her mother’s performance in letters of 1813 (29 January and 4 February).

Looking at more specific points: some of the extended quotations could be discussed rather than baldly left in the text. The translations seem rather stark and stilted, and would benefit from clearer integration given their interesting and pivotal role in the argument. Readers will need convincing that Austen’s approach stems from song rather than mostly from novel-reading. There could perhaps be some fuller discussion of the place of the Juvenilia (only mentioned on p. 7). If Austen’s initial attempts at novel-writing were as a virtuoso in the Juvenilia, this part of her expression survives to some extent in the letters but the published novels demonstrate her progression to performance as communication.

The assumption that Caroline Austen is a trustworthy authority can be challenged; she is involved in music during the last year of Austen’s life (Letters 23 January and 26 February 1817) and not just in the 1860s when her memory may have been inaccurate. Certainly the long quotation on page 10 needs to be justified and/or shortened. The section referencing Virginia Woolf should be cut so that the final focus is on the main subject of the discussion. Some minor quibbles about a reliance on the John Wiltshire book; references need to be curtailed/explained and perhaps referred to as ‘innovative’ or ‘well-argued’ rather than ‘wonderful’ (p. 6). What about the use of dashes in the letters? The 3rd edition of the letters is dated 1995 so probably a typo and there is an updated 4th edition of 2011. As references to the novels are only to P&P, the actual Oxford edition used could be cited. ‘Elizabeth’ on page 9 needs to be confirmed as either the Eliza under discussion or Edward Austen Knight’s wife (née Bridges). This would be a clear distinction (and helpful or otherwise for the Italian song thread) given the reportedly different relationships between Austen and these two sisters-in-law.

The topic is original but its scope needs to be more clearly defined as suggested. The writing style is clear but should adopt a more detached academic viewpoint without diversions into the personal. The article raises points of interest highly relevant to the special issue at which it is targeted but the argument would benefit from clearer focus and signposting using the advantage of the author’s access to specific material.

Reviewer 2 Report

I thoroughly enjoyed reading this paper, particularly the exploration of and connections between Austen the writer and Austen the musician. The paper is well supported with a range of relevant references and evidence sourced from Austen's novels (particularly Pride and Prejudice) and other sources (such as letters). The writing is generally clear and concise. There is a logical flow to the discussion and the insights are useful as well as interesting at times.

There are some opportunities here for strengthening the paper. Firstly, I suggest expanding the overview of the paper on lines 38-40. In particular, I suggest providing a more detailed outline of the discussion (such as the key points in their order) and then stating the overall argument of the paper. There is a useful point made in the conclusion about the musicality of Austen’s writing- this should definitely be foreshadowed in the introduction. Secondly, the paper provides quite a bit of detail about Pride and Prejudice and I wondered if Austen’s other novels could also be mentioned, perhaps briefly around line 128 (as a way to sum up the section and affirm the musicality of Pride and Prejudice as well as more widely in her writing). The third point that may assist with revision relates to the contribution of the paper to knowledge/scholarship on this topic. I suggest affirming how your discussion contributes to existing knowledge, perhaps in the introduction and/or conclusion more clearly and strongly. The fourth point to consider when revising the paper relates to the conclusion. Consider the relevance of the material here in your conclusion for the topic at hand (there may be an opportunity to trim out less relevant information and instead sum up your discussion in more detail). Lastly, there are some minor aspects of the written expression that can be attended to when you revise the paper (as below).

Lines 38-40 consider refinement of the phrasing to avoid any claims that you can determine Austen's attitudes (i.e. it is difficult to determine a person's attitudes without asking them directly unless you are careful in the phrasing used. For example, you could consider a revision of the phrasing to "Evidence from Austen's letters or personal writing suggest she had X attitude to musical and artistic virtuosity"). 

Line 44- Make it clear that you are referring to Pride and Prejudice

Lines 56 and 60 (and elsewhere in the paper)- please avoid starting a sentence with “so”

Lines 67-72- this is a very long sentence, consider splitting into 2 or 3 shorter sentences. Also references could be added for some of the ideas/points made here.

Lines 110-113- consider adding a reference for this point

Line 169 There seems to be an extra space after the quote

Line 237- there is “& and” in the sentence, perhaps this is a typo?

Line 295- consider replacing “looked” with a stronger verb such as investigated, explored

Line 381, 391 and 443- there seems to be an extra space on each of these lines

Line 439- use of “seem”, perhaps it should be “seemed” or “seems” depending on tense

Line 472- avoid contractions in academic writing so replace “it’s” with “it is”

Line 497-500- is this a quote? It is introduced as a quote but not formatted as a quote as you have done elsewhere.   
